# Short-term Source Apportionment of Fine Particulate Matter with Time-dependent Profiles Using SoFi: Exploring the Reliability of Rolling Positive Matrix Factorization (PMF) Applied to Bihourly Molecular and Elemental Tracer Data

Qiongqiong Wang[1,2], Shuhui Zhu[3], Shan Wang[4], Cheng Huang[3], Yusen Duan[5], Jian Zhen Yu[2,4]

[1] Department of Atmospheric Science, School of Environmental Studies, China University of Geosciences, Wuhan, China.
[2] Department of Chemistry, The Hong Kong University of Science & Technology, Hong Kong, China.
[3] State Environmental Protection Key Laboratory of the Cause and Prevention of Urban Air Pollution Complex, Shanghai Academy of Environmental Sciences, Shanghai, China.
[4] Division of Environment & Sustainability, The Hong Kong University of Science & Technology, Hong Kong, China.
[5] Shanghai Environmental Monitoring Center, Sanjiang Road, Xuhui District, Shanghai, China

*Correspondence to*: Jian Zhen Yu (jian.yu@ust.hk)

**Abstract.** Positive matrix factorization (PMF) has been widely used to apportion the sources of fine particulate matter ($PM_{2.5}$) by utilizing PM chemical speciation data measured at receptor site(s). Traditional PMF, which typically relies on long-term observational datasets of daily or lower time resolution to meet the required sample size, has its reliability undermined by changes in source profiles, thus it is inherently ill-suited for apportioning sporadic sources or ephemeral pollution events. In this study, we explored short-term source apportionment of $PM_{2.5}$ using a set of bihourly chemical speciation data over a period of thirty-seven days in the winter of 2019-2020. PMF run with campaign-wide data as input ($PMF_{ref}$) was initially conducted to obtain reference profiles for the primary source factors. Subsequently, short-term PMF analysis was performed using the Source Finder Professional (SoFi Pro). The analysis sets a window length of 18 d and constrained the primary source profiles using the *a*-value approach embedded in SoFi software. Rolling PMF was then conducted with a fixed window length of 18 d and a step of 1 d using the remaining dataset. By applying the *a*-value constraints to the primary sources, the rolling PMF effectively reproduced the individual primary sources, as evidenced by the slope values close to unity (i.e., 0.9-1.0). However, the estimation for the firework emission factor in the rolling PMF was lower compared with the $PMF_{ref}$ (slope: 0.8). These results suggest the unique advantage of short-term PMF analysis in accurately apportioning sporadic sources. Although the total secondary sources were well-modelled (slope: 1.0), larger biases were observed for individual secondary sources. The variation in source profiles indicated higher variabilities for the secondary sources, with average relative differences ranging from 42% to 173%, while the primary source profiles exhibited much smaller variabilities (relative differences of 8-26%). This study suggests that short-term PMF analysis with the *a*-value constraints in SoFi can be utilized to apportion primary sources accurately, while future efforts are needed to improve the prediction of individual secondary sources. Additionally, future rapid source apportionment analysis can benefit from utilizing a library of source profiles derived from existing

measurement data, thereby significantly reducing the time lag associated with receptor modelling source apportionment techniques.

## 1 Introduction

Atmospheric particulate matter with aerodynamic diameter less than 2.5 μm ($PM_{2.5}$) is known to negatively impact human health and exert noticeable but highly uncertain effect on climate change (IPCC, 2014). Epidemiological studies have consistently demonstrated that exposure to $PM_{2.5}$ can result in various cardiovascular and chronic respiratory diseases (Yin et al., 2020). The implementation of stringent control measures since 2013 has led to declining concentrations of $PM_{2.5}$ in many megacities in China, with annual-average decreased from 72.3 μg m$^{-3}$ in 2013 to 47.4 μg m$^{-3}$ in 2017, as calculated from

monitoring data in 74 cities across China (Wang et al., 2020; Chow et al., 2022a). However, the annual-mean PM level in many cities remain above the new WHO guideline (5 μg m$^{-3}$) by a large margin. As importantly, short-term pollution episodes continue to occur frequently in recent years (e.g., Shao et al., 2018; Wang et al., 2022). Recognizing the need to reduce the severity and frequency of episodic pollution incidents, it becomes evident that achieving episode-scale source apportionment is essential.

Receptor models such as positive matrix factorization (PMF) and chemical mass balance (CMB) have been widely deployed to apportion the sources of $PM_{2.5}$ based on observation-based composition data (Paatero and Tapper, 1994; Watson et al., 1984). The CMB model can apportion the source contributions of a single sample in principle, but the uncertainties can be large due to the high variability in the source profiles (Lee and Russell, 2007) as the local-specific profiles are often unavailable in many places. While the PMF model has the advantage of avoiding the need to input source profiles, it requires a large

sample size to do the source apportionment. PMF assumes constant source profiles throughout the entire sampling period (Reff et al., 2007). Due to the limited time resolution from offline filter-based sampling schedule, e.g., sampling duration of 24 h and sampling frequency of once every three or six days, PMF is often conducted using the Environmental Protection Agency-EPA PMF software (Norris et al., 2014) with data spanning one or multi years to meet the sample size requirement (e.g., Chow et al., 2022b; Scotto et al., 2021). As a result, there is a notable time lag in obtaining the source apportionment results and

implementing relevant policy controls. There is an urgent need for rapid source apportionment methods that can provide timely policy implications.

Source profile changes are often expected over an extended period of observation for certain sources. For instance, biomass burning exhibits variations in dominant biomass materials during different seasons; the implementation of catalytic converter replacement program alters the source profiles of vehicular emissions (Lee et al., 2017). Sporadic sources, such as firework

emission during holidays or wildfires during dry seasons, can significantly contribute to PM pollution episodes that persist for hours to days, often overshadowing the effects of reductions in anthropogenic emissions (Song et al., 2021; Kong et al., 2015). The PMF analysis using long-term data sets could not properly reflect source profile changes experienced during the long-time span. In other words, long-term PMF is inherently unsuitable for apportioning sporadic sources or ephemeral pollution

events. This limitation explains the common observation that PMF with robust mode tends to underestimate the high concentration data while overestimating the low concentration data (Henry and Christensen, 2010). Consequently, contribution estimates of these sources would be biased when apportioned alongside other regular sources using long-term observational data.

By implementing online measurement techniques, researchers are able to conduct source apportionments studies based on hourly PM chemical speciation data covering several weeks to months. Such studies can circumvent the issue of source profile changes arising from the long-term sampling (Wang et al., 2018). Recently, Canonaco et al. (2021) introduces a new method called "rolling PMF" to conduct the source apportionment with time-dependent source profiles using the SoFi software. In this method, PMF is performed over a small, moving time frame (e.g., a window length of 2 weeks with step of 1 day), allowing that the factor profiles to evolve with time. To decrease the rotational ambiguity, short-term rolling PMF is conducted with the source profile constraints using the $a$-value approach embedded in the SoFi program (Canonaco et al., 2013). This method has been demonstrated using the one- and multi-year non-refractory sub−micrometer aerosol chemical speciation monitor (ACSM) dataset (Canonaco et al., 2021; Chen et al., 2021, 2022) for the source apportionment of organic aerosols (OA). The source profiles of primary factors obtained from the traditional PMF runs conducted in each season are selected as the reference profiles. With the source profile constraints, the rolling PMF can effectively capture the individual primary organic aerosol (POA) source and total oxygenated organic aerosol (OOA) sources when compared with the traditional PMF. However, noticeable differences in individual OOA sources were observed. The rolling PMF (or moving window PMF) method has been also applied to the hourly $PM_{2.5}$ chemical speciation data measured in Tianjin during a two-month field campaign, including ions, organic carbon (OC), elemental carbon (EC) and elements (Song et al., 2021), where PMF runs were performed without the source profile constraints using EPA PMF software. The apportioned results, without the source-specific organic tracers, showed clear mixing of several source factors in Song et al. (2021). The application of the rolling PMF method with time-dependent source profile constraints holds potential for rapid source apportionment when source profiles are available from existing chemical speciation measurement data.

A comprehensive online measurement campaign was conducted at a suburban site in Shanghai during a period of 37 days in the winter of 2019-2020 (specifically from 29 Dec. 2019 to 9 Feb. 2020), encompassing both the pre-lockdown and lockdown phases of the Covid-19 pandemic. This data collection effort involved hourly measurements of major ions, OC, EC, elements, as well as bihourly measurements of source-specific organic tracers in $PM_{2.5}$. Notably, this time frame captured the dynamic changes in pollution sources and included a sporadic source event—firework emissions during the Chinese New Year (CNY) and Lantern festival. Thus, it presented a unique opportunity to evaluate a shot-term PMF strategy. A thorough traditional source apportionment analysis conducted using the EPA PMF software is documented in our previous study (Wang et al., 2022b). In this study, we specifically investigated the applicability of a short-term source apportionment strategy using the bihourly $PM_{2.5}$ chemical speciation data with the SoFi software and compared with those obtained through the traditional PMF. The findings of this study offer valuable insights into the future development of rapid source apportionment methods for $PM_{2.5}$,

particularly for short-term periods and episodic events. These insights have the potential to enhance air quality management practices.

## 2 Methods

### 2.1 Sampling and chemical analysis

The field campaign was conducted during 29 Dec. 2019 to 9 Feb. 2020 at Dianshan Lake (DSL) supersite (31.09°N, 120.98°E) in Shanghai, China. The sampling site was located in a suburban area, about 50 km away from downtown Shanghai and with relatively low influences of local anthropogenic sources. PM speciation measurements included hourly major ions (sulfate, nitrate, and ammonium) by a Monitoring AeRrosols and Gases in ambient Air system (MARGA), OC and EC by a Sunset Semi-Continuous Carbon Analyzer, elements (i.e., K, Ca, Cr, Mn, Fe, Cu, Zn, As, Se, Ba, and Pb) by an energy dispersive X-ray fluorescence spectrometer (XRF) and bihourly organic tracers (hopanes, steranes, levoglucosan, mannosan, phthalic acid, 2,3-dihydroxy-4-oxopentanoic acid (DHOPA), β-caryophyllinic acid (β-caryT) and α-pinene secondary organic aerosol (SOA) tracers (α-pinT)) by a Thermal desorption Aerosol Gas chromatography–mass spectrometry (TAG). TAG data during 16-21 Jan. 2020 were not available due to instrument maintenance. For detailed information about the sampling site and chemical analysis procedures, refer to our previous paper (Wang et al., 2022b).

### 2.2 Source apportionment

In this work, positive matrix factorization (PMF) with the multilinear engine version 2 (ME-2) (Paatero, 1999) in the interface of SoFi Pro (version 8) (Canonaco et al., 2013) was adopted to apportion the sources contributing to $PM_{2.5}$ mass. The PMF model in matrix notation is defined as Eq. (1-2)

$$x_{ij} = \sum_{k=1}^{p} g_{ik}f_{kj} + e_{ij} \qquad (1)$$

$$Q = \sum_{i=1}^{n}\sum_{j=1}^{m}\left[\frac{e_{ij}}{u_{ij}}\right]^{2} \qquad (2)$$

where $x_{ij}$ is the measured concentration, $n$ is the number of samples, $m$ is the number of species, $p$ is the number of factors, $g_{ik}$ is the source contribution of the $k^{th}$ factor to the $i^{th}$ sample, $f_{kj}$ is the factor profile of $j^{th}$ species in the $k^{th}$ factor, $e_{ij}$ is the residual of $j^{th}$ species in $i^{th}$ sample and $u_{ij}$ is the user-defined uncertainty. $Q$ is the objective function representing the uncertainty weighted difference between observed and modeled species concentrations. PMF finds the final solution by minimizing the $Q$ value.

Factor analysis methods like PMF are known to encounter rotational ambiguity, whereby different combinations of source contribution G and source profile F matrix can yield the same $Q$ value. This issue often results in mixed factors or environmentally unrealistic factors. Previous studies have demonstrated the effectiveness of constraining expected source profiles using the *a*-value approach embedded in SoFi software (Canonaco et al., 2013). The *a*-value approach allows for the

imposition of constraints on the source profiles/contributions from the given reference profiles/contributions, with a certain degree of variation from the anchoring profiles (Eq 3-4).

$$f'_{kj} = f_{kj} + a \times f_{kj} \qquad (3)$$

$$g'_{ik} = g_{ik} + a \times g_{ik} \qquad (4)$$

130 Here, the index $j$, which varies between 0 and the number of species-$m$, represents the species of the $k^{th}$ factor. The index $i$, which varies between 0 and the number of samples-$n$, is the sample of the $k^{th}$ factor. $f_{kj}$ and $g_{ik}$ are the anchoring profiles and anchoring contributions, respectively, while $f'_{kj}$ and $g'_{ik}$ are the output source profiles and source contributions, respectively. The scalar $a$ ranges from 0 to 1, which determines the extent to which the output $f'_{kj}$ / $g'_{ik}$ is allowed to vary from the input reference $f_{kj}$ / $g_{ik}$. For example, a $a$ value of 0.3 corresponds to 30% variation, while a $a$ value of 1 is equivalent to a

135 completely unconstrained (or free) PMF situation.

Figure 1 illustrates the flowchart outlining the source apportionment methodology employed in this study. Initially, a PMF run was conducted by EPA PMF software using campaign-wide bihourly data as input (referred to as PMF$_{ref}$) to derive the reference profiles for primary sources. Subsequently, the first sampling period data of 18 d was utilized to perform the short-term PMF run and evaluate the effectiveness of the $a$-value approach using SoFi. The source profiles obtained in PMF$_{ref}$ was

140 used as the reference profiles in the $a$-value approach to help PMF find the environmentally reasonable solution. Following this, the rolling PMF was conducted using the remaining dataset with the optimum $a$ values to validate the short-term PMF results. The results of the rolling PMF analysis were discussed and compared with the results obtained from PMF$_{ref}$. The 22 input species for both the PMF$_{ref}$ and the short-term rolling PMF runs include sulfate, nitrate, ammonium, OC, EC, K, Ca, Mn, Fe, Cu, Zn, As, Ba, Pb, and 8 organic species (hopanes, steranes, levoglucosan, mannosan, phthalic acid, α-pinT, β-caryT, and

145 DHOPA). The specific input data utilized in individual PMF runs are shown in Table S1.

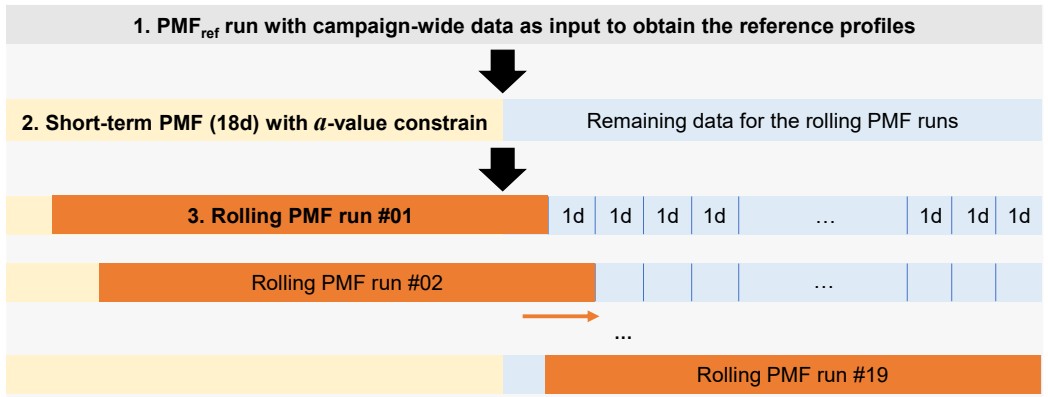

**Figure 1. Flow diagram of the short-term PMF strategy used in this study.**

# 3 Results and discussion

## 3.1 Overview of the PM pollution at DSL site

Figure 2a shows the temporal variation of $PM_{2.5}$ and select tracers during the campaign period, with the average concentrations provided in Table S2. The sampling period was divided into two distinct sub-periods: (1) before CNY (29/12/2019-23/1/2020) and (2) CNY and post-CNY (24/1-9/2/2020). The CNY (25 Jan.) and Lantern festival (8 Feb.) fell within the second period when the lockdown restriction had been implemented. A clear reduction of the concentrations for most tracer species was observed during the CNY and post-CNY period, except for K and Ba (Text S1 and Figure 2a). It is known that combustion of fireworks emits particles enriched with elements such as Sr, K, Ba, Cu, Bi and etc. (Manousakas et al., 2022). Scatter plots of measured K with the source tracers-levoglucosan from biomass burning and Pb from coal combustion unequivocally indicated the presence of firework emission source during the CNY holiday and Lantern festival (Figure 2b). The combustion of fireworks during these events led to significantly elevated K concentrations. Conversely, during the remaining time period, K primarily originated from biomass burning and coal combustion, as evidenced by the strong correlation with the corresponding tracer species.

Source apportionment results over entire sampling period (i.e., $PMF_{ref}$) supplies an overview about the emission sources at this site. A thorough source apportionment result for this site can be found in our previous paper (Wang et al., 2022b), where 14 factors were resolved using a list of more comprehensive input species over the entire sampling period. Among these factors, the PAH-rich factor, cooking emission, and one SOA factor are negligible $PM_{2.5}$ contributors (<1%). The contribution of the residual oil combustion factor to $PM_{2.5}$ is also minor (<3%). Additionally, the detection frequency of V, a tracer for the residual oil combustion factor, was lower than 50% for the short-term input time window. Thus, these four factors were not incorporated in this study, and we focus on the 10 major factors resolved in our $PMF_{ref}$ run. Given the limited data points available for the short-term PMF runs, this approach allows us to obtain a more robust solution, aligning with the study's objective of testing the short-term PMF strategy. The robustness of the $PMF_{ref}$ result was tested by the bootstrap and displacement error estimation method embedded in EPA PMF 5.0 software (Norris et al., 2014). All bootstrap factors mapped to the base factors in >95% of the runs. No factor swaps and no decrease of $Q$ were observed in DISP. The PMF-modelled reconstructed $PM_{2.5}$ mass is close to the measured one, with slope of 1.01 and $R_p$ of 0.99. The model performance for individual species was also good, with slopes ranging from 0.59 to 1.08 and $R_p$ in the range of 0.82-1.00.

Briefly, the $PMF_{ref}$ run resolved 10 factors comprises four secondary sources (i.e., secondary nitrate formation process, secondary sulfate formation process, and two SOA factors-SOA_I and SOA_II) and six primary factors (i.e., vehicle exhaust, industrial emissions, coal combustion, dust, biomass burning, and firework emissions). The SOA_I factor contained high loadings of α-pinene and toluene SOA tracers, representing mixture of biogenic and anthropogenic SOA. The SOA_II factor was primarily contributed by phthalic acid, suggesting an anthropogenic origin. Among the primary factors, the firework emission factor was only present during the CNY and post-CNY sampling period (Figure 2b). Consequently, we imposed constraints to set the factor contributions of firework emissions to zero during the period before CNY. The resolved factor

profiles and PM$_{2.5}$ contributions from PMF$_{ref}$ are shown in Figure S2 and Figure S3. Briefly, the PMF$_{ref}$ results showed that secondary nitrate and secondary sulfate factors constituted the most important sources contributing to the PM$_{2.5}$ levels at this site, accounting for 58% and 11% of the PM$_{2.5}$ mass during the period before CNY, and 40% and 23% during the CNY and post-CNY period, respectively. SOA_I and SOA_II contributed to 3% and 7-8% of the PM$_{2.5}$ mass, respectively. Among the primary sources, industrial emissions, biomass burning, and dust showed comparable contributions to the PM$_{2.5}$ mass (ranging from 2% to 8%), while vehicle exhaust was a minor source, contributing less than 1% to PM$_{2.5}$ at this suburban site. Firework emissions, however, constituted a non-negligible source during the CNY and post-CNY period, contributing to 12% of the total PM$_{2.5}$ mass.

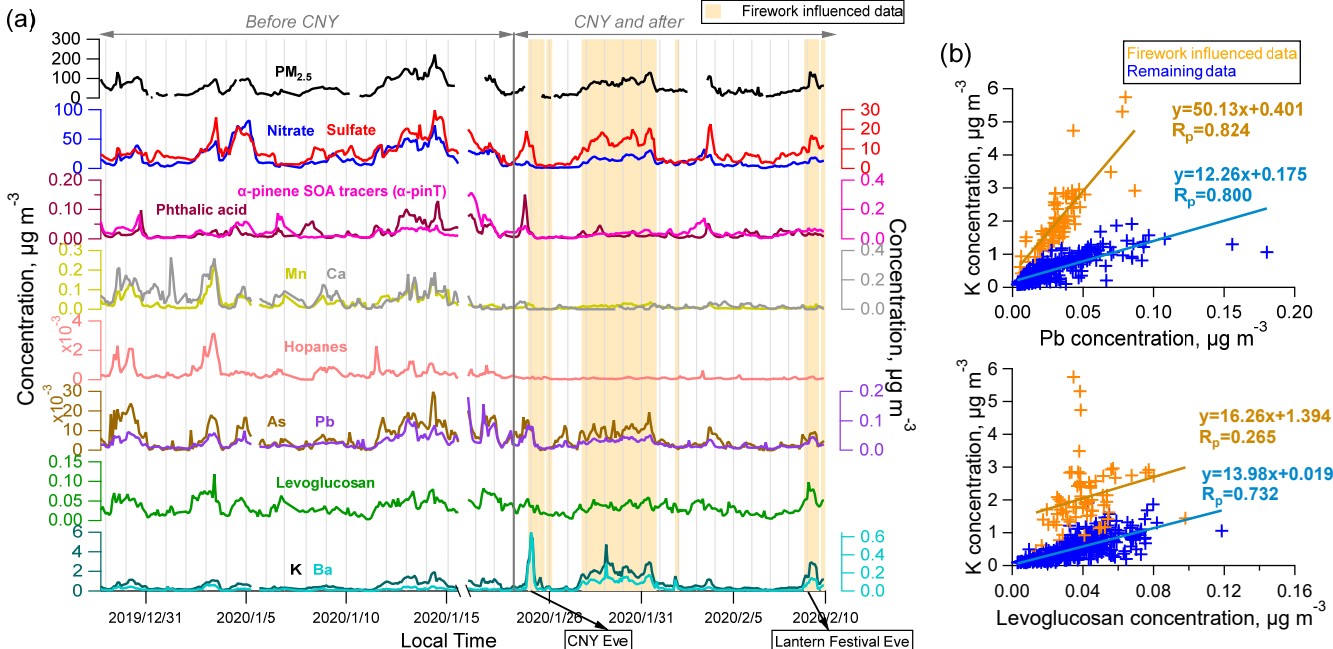

Figure 2. (a) Time series of concentrations of total PM$_{2.5}$ and selected tracer species from 29 Dec. 2019 to 9 Feb. 2020 at the DSL site in Shanghai. The campaign period was divided into two phases: before Chinese New Year (CNY) and CNY and post-CNY period. The data influenced by firework emissions are highlighted in light orange. (b) Scatter plots illustrating the relationship between K concentrations and two other tracer species, Pb and levoglucosan, during the firework-influenced period and the remaining period.

### 3.2 Short-term PMF run combined with *a*-value approach

The short-term source apportionment analysis was conducted using data from the first sampling period, spanning 18 days from 29 Dec. 2019 to 15 Jan. 2020. The selection of the window length may vary depending on the specific data sets under study. The determination of the window length for our observational data set is shown in Text S2, where 4 d, 7 d, 10 d, 14 d and 18 d were initially evaluated. A window length of 18 d was chosen as it produced the most stable base run result with minimum factor profile mixing. Previous studies that employed higher time resolution measurements (e.g., hourly or 30-min intervals)

suggested a window length of 14 d (Chen et al., 2022; Canonaco et al., 2021; Song et al., 2021). However, our bihourly time-resolution data indicated a slightly longer window length, which provided a more robust solution.

The short-term PMF run resolved nine factors, with the firework emission factor not resolved during the sampling period before CNY. The $a$-value approach was tested in the short-term PMF run, utilizing the source profiles of primary factors obtained from $PMF_{ref}$ as reference profiles. A range of $a$ value, from 0 to 1 with a step size of 0.1, was tested. Compared with primary sources, the secondary sources often do not represent specific emissions. Instead, they typically result from a complex interplay of multiple aging processes that occur over the observational period and are susceptible to environmental conditions such as relative humidity and photoactivity, etc. Thus, the four secondary factors were not constrained in the short-term PMF run using the $a$-value approach, consistent with the common strategy in previous studies (Chen et al., 2022; Canonaco et al., 2021). A total of 100 PMF calls were performed and the variability of the $Q/Q_{exp}$ was examined. The ratio $Q/Q_{exp}$, where $Q_{exp}$ $\approx n \times m - p \times (n + m)$, indicates the overall fitting of all input species and is reciprocally associated with the fitting (Norris et al., 2014). Among the 100 runs, the variation of the $Q/Q_{exp}$ are consistently minimum, with a coefficient of variation of <1%. The one with the lowest $Q/Q_{exp}$ was chosen for further analysis. For comparison, unconstrained PMF run was also conducted in a similar manner. In general, the $a$-value constrained PMF runs showed better agreement with the $PMF_{ref}$ compared to the unconstrained PMF run (Figure S5a). The change in $Q/Q_{exp}$ values was evaluated to determine the optimum $a$ values (Figure S5b). Larger $Q/Q_{exp}$ values were observed in the $a$-value constrained runs, compared with the unconstrained PMF run. As the $a$ values decreased from 1 to 0, the $Q/Q_{exp}$ increased, reflecting a decrease in the freedom of the source profiles. The change in $Q/Q_{exp}$ exhibited a "U" shape, with higher values observed for small (0-0.2) and large $a$ values (0.9-1), indicating larger changes in the PMF results with varying $a$ values. A threshold $a$ value of 0.3 was initially selected, after which the change in $Q/Q_{exp}$ became considerably smaller.

Figure 3 presents a comparison of the relative difference in $PM_{2.5}$ source contributions for individual primary source factors obtained from the $a$-value constrained runs and the unconstrained PMF run, in relation to the $PMF_{ref}$ results. Different factors showed different response to the change in the $a$ values. For vehicle exhaust, industrial emissions, and coal combustion, much smaller differences (0-15%) were observed with small $a$ values (<0.5). However, as the $a$ values increased, the differences became more substantial (10-60%), highlighting the importance of constraining the source profiles for these factors. In the case of dust and biomass burning, larger differences were observed (22-44% and 10-21%, respectively) when the $a$ values exceeded 0.1. Therefore, smaller $a$ values were suggested for the two sources, which was in accordance with the fact that their source profiles were less affected by lockdown restrictions compared to other primary sources. After initial test, an $a$ value of 0.1 was selected for biomass burning and dust, while an $a$ value of 0.3 remained for other primary factors. These chosen $a$ values (0.1 and 0.3) align with previous studies that utilized ACSM datasets, where $a$ values between 0-0.4 were adopted (Canonaco et al., 2021). With this set of optimized $a$ values, the relative differences in the apportioned $PM_{2.5}$ source contributions compared to those apportioned by $PMF_{ref}$ were as follows for the five primary factors: vehicle exhaust (-1%), industrial emission (-11%), coal combustion (5%), dust (-14%), and biomass burning (-5%). In comparison, the unconstrained PMF produced notably poorer results for vehicle exhaust (35%) and biomass burning (-17%).

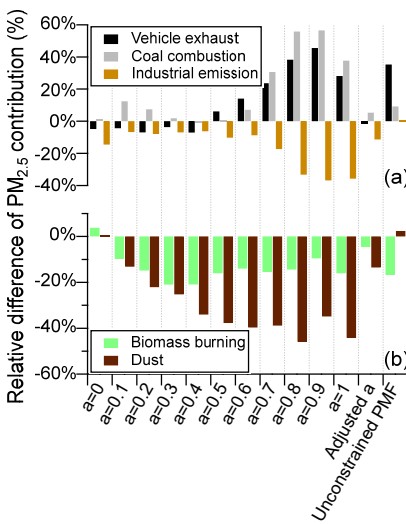

**Figure 3. Relative differences in PM$_{2.5}$ contribution between different *a*-value constrained runs and the unconstrained PMF run, compared to the reference result, for (a) vehicle exhaust, coal combustion, and industrial emission, and (b) biomass burning and dust. The "adjusted *a*" indicated the final *a* values adopted, i.e., *a*=0.3 for vehicle exhaust, coal combustion, and industrial emissions, and *a*=0.1 for biomass burning and dust.**

We additionally conducted a sensitivity test on the reference profiles by manually generating a set of new reference profiles that deviated from the original profiles by a relative standard deviation ranging from 10% to 70%. The details are shown in Text S3. As the deviation increased, the apportioned source contributions exhibited greater relative differences compared to PMF$_{ref}$ for the primary factors (Figure S6). These findings indicate that utilizing source profiles derived from PMF$_{ref}$ is an effective approach for establishing appropriate constraints, resulting in a closer approximation to the true source profiles at the site.

## 3.3 Short-term rolling PMF runs combined with *a*-value approach

We next tested whether the short-term PMF strategy works on more dataset with potential change of pollutions. The rolling PMF runs (denoted as $PMF_{roll}$) were conducted using the remaining dataset, maintaining a fixed window length of 18 d. The window increment was set at 1 d, following the practice in previous studies (Canonaco et al., 2021; Chen et al., 2021, 2022; Song et al., 2021). A total of 19 $PMF_{roll}$ runs were performed (Table S1). The first two $PMF_{roll}$ runs utilized input data collected before the CNY (30 Dec. 2019-23 Jan. 2020) and resolved nine factors. Subsequently, the 3[rd] to 19[th] $PMF_{roll}$ runs employed input data spanning the CNY (1 Jan.-9 Feb. 2020) and resolved ten factors, including an additional factor attributed to firework emissions. The 3[rd] $PMF_{roll}$ run (input data of 1-24 Jan. 2020), a transitional PMF run from 9 to 10 factor, was excluded due to the limited availability of data points influenced by firework emissions ($N_{firework\_data}$=2, representing the number of data points under the influence of firework emissions). Furthermore, the apportioned results from this run displayed significant discrepancies compared with the rest of the $PMF_{roll}$ runs. Consequently, 18 out of the 19 $PMF_{roll}$ runs were selected for further analysis.

Figure 4 shows the time series of the $PM_{2.5}$ source contributions from individual $PMF_{roll}$ runs and the average contributions. Comparable results were observed across the $PMF_{roll}$ runs for all primary source factors, indicating the effectiveness of the *a*-value approach to reproduce the primary source contributions during the short-term PMF runs. To illustrate this point, we also performed unconstrained rolling PMF runs (i.e., without the *a*-value approach), which showed much larger run-to-run variability for the primary source factors, especially vehicle exhaust and coal combustion (Text S4 and Figure S7). These findings underscore the advantage of employing source profile constraints to achieve reproducible source apportionment results when performing the PMF analysis over a short-term measurement period. The four secondary source factors were not subject to constraints and displayed varying levels of run-to-run variability. Secondary nitrate exhibited minimal variability among the runs, while secondary sulfate showed larger variations. Both SOA factors demonstrated even greater variations, particularly the SOA_I factor. However, the SOA_II factor exhibited relatively smaller variations in the later sampling period data.

The final solution was obtained by averaging the $PM_{2.5}$ source contributions from all $PMF_{roll}$ runs, which were then compared with the reference result obtained from $PMF_{ref}$ (Figure 5). The primary source factors (i.e., vehicle exhaust, industrial emission, coal combustion, and dust) exhibited a strong agreement between the $PMF_{roll}$ and $PMF_{ref}$ results (slope>0.93). A slight underestimation was observed for biomass burning, with a slope of 0.90. In contrast, the sporadic source of firework emissions showed consistently lower estimations by $PMF_{roll}$ (slope 0.81), which may reflect higher source contributions by $PMF_{ref}$. This result highlights the unique advantage of the short-term source apportionment in accurately apportioning the sporadic sources (Song et al., 2021). Among the four secondary sources, secondary nitrate showed good agreement with the reference result (slope of 1.0 and Pearson correlation coefficient-$R_p$ of 1.0). Secondary sulfate exhibited a good correlation with the $PMF_{ref}$ (slope=1.2 and $R_p$=0.92), although the $PMF_{roll}$ runs apportioned higher contributions, especially for the later sampling period during the lockdown. SOA_I showed a weaker correlation with the reference result ($R_p$=0.77), and the slope varied with time

(Figure 5). On the other hand, SOA_II displayed good agreement between PMF$_{roll}$ and PMF$_{ref}$, but larger uncertainties were associated with the apportioned results due to large run-to-run variabilities observed in the source contributions, especially

during the middle sampling period (Figure 4). Notably, the sum of the four secondary sources showed good agreement with the PMF$_{ref}$, both with (slope=1.0 and R$_p$=1.0) and without *a*-value constraints (slope=0.95 and R$_p$=0.99). This observation may be attributed to the intrinsic temporal variations differing between primary and secondary sources.

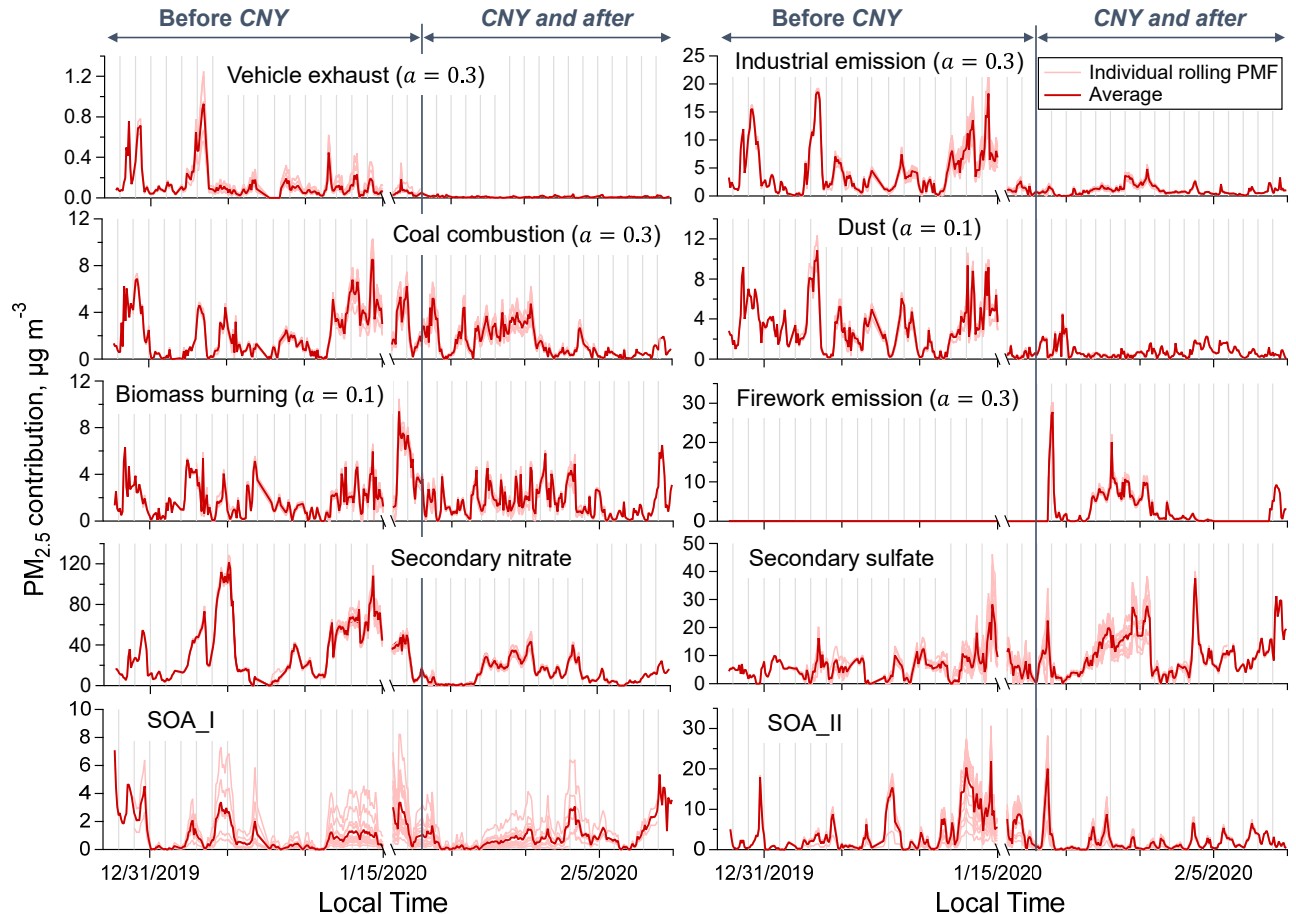

**Figure 4. Time series of factor contributions to PM$_{2.5}$ for individual PMF$_{roll}$ runs and the average source contributions. The**
**individual PMF$_{roll}$ run is shown in light red line and the average PMF$_{roll}$ result is shown in dark red line.**

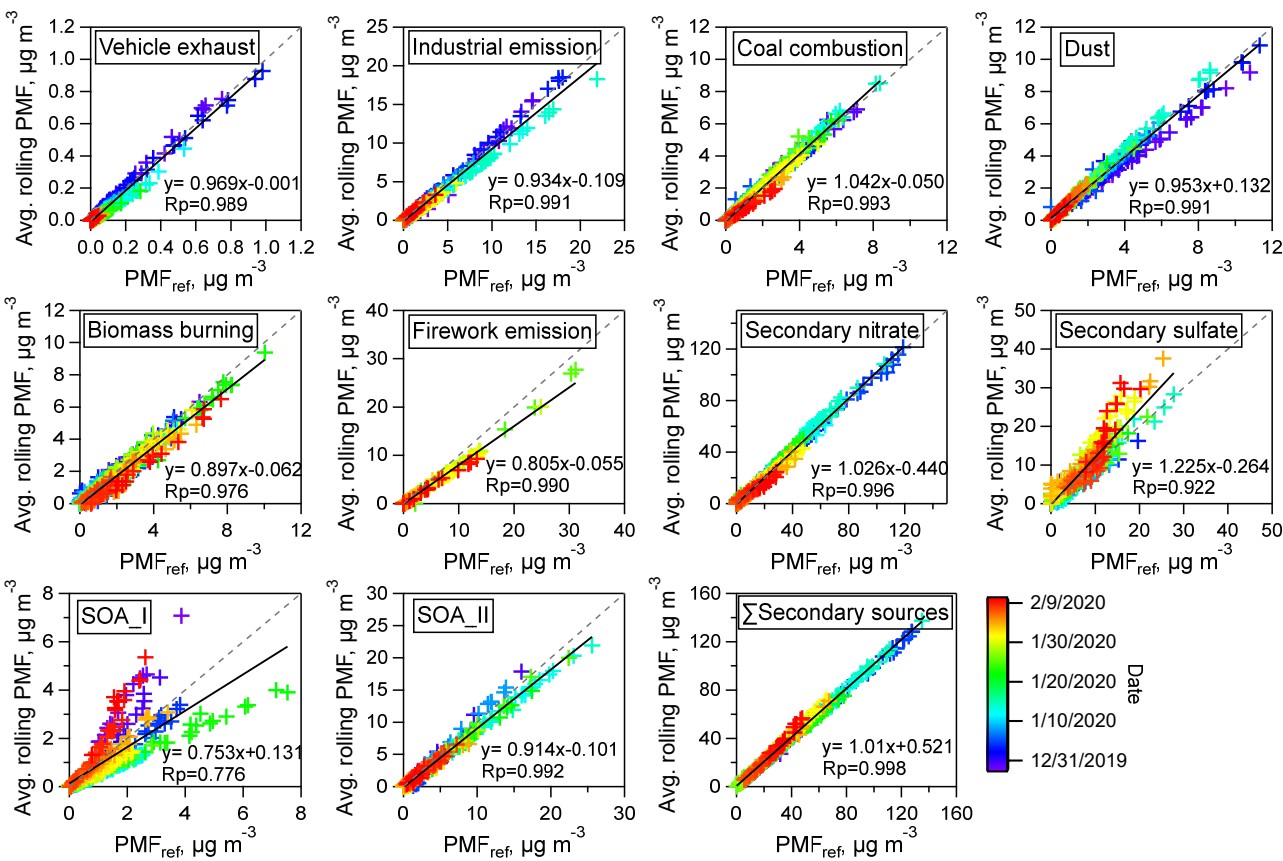

**Figure 5. Comparison of the PM₂.₅ source contributions obtained from average PMF_roll runs with the reference result in PMF_ref for individual source factors and the sum of the four secondary sources.**

### 3.4 Source profile variability

The temporal variation of source profiles is the fundamental reason why short-term source apportionment is necessary to achieve accurate source apportionment during episodic events. Figure 6 presents the average factor profiles of the ten resolved source factors throughout the entire field campaign, alongside the reference profiles from PMF_ref. The error bars represent one standard deviation of profile variability across the PMF_roll runs throughout the entire measurement period. This variability encompasses both time-dependent variations in the factor profiles and uncertainties associated with the PMF analysis. All

primary factors showed comparable source profiles between PMF_roll and PMF_ref. However, the four secondary source factors derived from PMF_roll showed higher variabilities in their profiles and larger differences compared to the PMF_ref. In particular, the secondary nitrate and sulfate factors from PMF_roll showed higher loadings of organic tracers and elemental species in their profiles compared to PMF_ref. The SOA_I factor showed a higher proportion of inorganic ions, whereas the SOA_II factor showed lower loadings of the inorganic ions.

We calculated the relative difference between the source profiles obtained from $PMF_{roll}$ with $PMF_{ref}$ to evaluate their disparities (Figure 7). The relative difference for each $PMF_{roll}$ run was calculated as the average value of the relative difference for all input species. The results indicated that the primary sources showed relatively small differences among individual $PMF_{roll}$ runs. For example, the relative difference for vehicle exhaust varied from 17% to 33%. Across the five primary factors, the average relative difference ranged from 8% for dust and biomass burning to 26% for vehicle exhaust. In contrast, the secondary sources

inherently displayed more variability than the primary sources, leading to challenges and larger uncertainties in apportioning individual secondary sources. Significant variabilities were observed in the source profiles of the secondary sources. Among them, secondary sulfate showed slightly smaller relative difference, with an average value of 42% (range 26-60%). Secondary nitrate, SOA_I and SOA_II showed large variations, with an average relative difference of 173%, 162%, and 75%, respectively. In the case of secondary nitrate factor, although the apportioned $PM_{2.5}$ contributions from individual source factors were

comparable to the reference result, the resolved source profiles exhibited high time-dependent variabilities. We hypothesize this may be attributed to the sensitivity of nitrate formation to the reduction of $NO_x$ and VOC precursors during the lockdown restriction (Yang et al., 2022). Previous laboratory studies indicated that reducing anthropogenic pollutants such as $SO_2$ and $NO_x$ can also reduce the biogenic SOA formation via anthropogenic–biogenic interactions (Zhang et al., 2019; Xu et al., 2015). This, to some extent, explains the high variabilities in source profiles of the two SOA factors. Additionally, the high

variabilities may also arise from the uncertainties in the PMF analysis due to the limited data points available from the short-term time span (Wang et al., 2018). Therefore, in future studies, alternative approaches are needed to independently assess the contribution of secondary sources. Also, we recommend deploying higher time resolution measurement of the organic tracers. This will help ensure accurate source apportionment results for individual secondary sources, especially within the confines of a short-term time span.

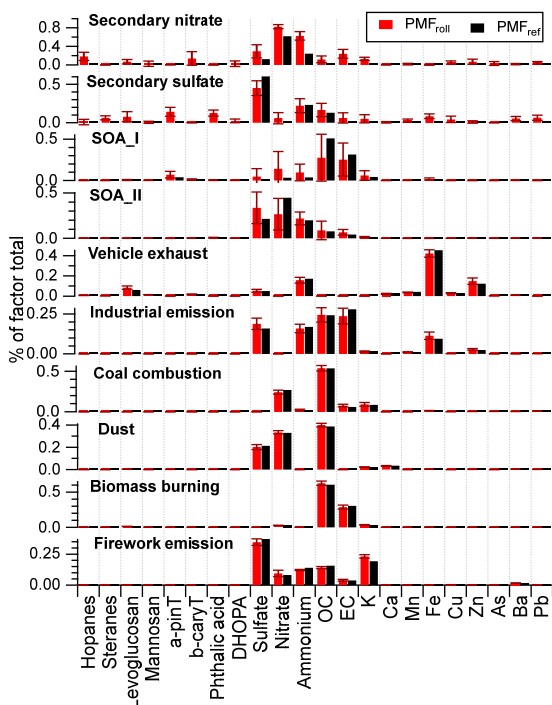


**Figure 6. Comparison of the source profiles (% of factor total) derived from the short-term rolling PMF runs (PMF_roll) and the reference profiles from PMF_ref. Error bars represent one standard deviation of profile variability across the PMF_roll runs.**

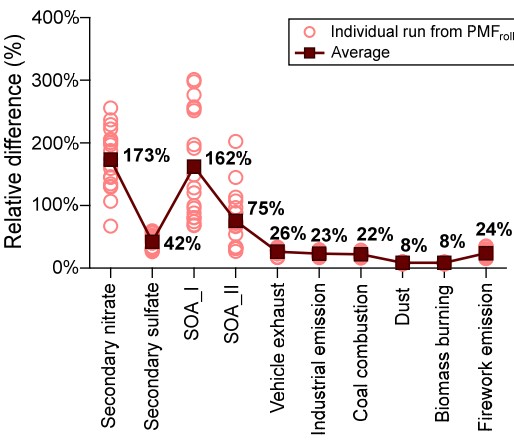


**Figure 7. The relative difference in the resolved source profiles among the individual rolling PMF runs. The relative difference for individual rolling PMF run (empty circles) was calculated as the average value of the relative difference for all input species. Solid squares represent the average value from all rolling PMF runs.**

## 4 Conclusions

In this study, we presented a short-term PMF strategy utilizing bihourly PM chemical speciation data including the molecular and elemental tracers. Initially, the PMF_ref using the campaign-wide measurement data were performed by EPA PMF software

to get an overview of the emission sources and obtain the reference profiles of the primary sources. Then, the short-term PMF analysis was performed using an 18-day window length combined with the *a*-value approach in SoFi software. The reference profiles derived from the campaign-wide data were employed as constraints to reduce the rotational ambiguity in the short-term PMF results. The training data with the *a*-value constraints for an 18-day window indicated a smaller "*a* value" for biomass burning and dust sources. This suggests that the profiles of these sources remain relatively constant and exhibit less variability throughout the campaign period. The constrained PMF results exhibited improved agreement with the reference results compared to the base run without any constraints. The rolling PMF analysis with optimized *a*-value constraints demonstrated good agreement between the regular primary sources and the reference result, underscoring the efficacy of source profile constraints in short-term PMF runs. However, the sporadic source of firework emissions exhibited overestimation in the long-term source apportionment results. Furthermore, noticeable differences were observed between the rolling PMF and $PMF_{ref}$ for individual secondary sources, particularly the SOA factors. Nevertheless, the overall contribution of the total secondary sources showed good agreement. Future endeavors should target to improve the modelling of individual secondary factors by either using alternative approaches or deploying higher time resolution measurement of organic tracers.

The findings of this study highlight the applicability of the short-term PMF analysis with source profile constraints for source apportionment of $PM_{2.5}$. This suggests the potential for future work to achieve rapid source apportionment by utilizing a library of source profiles derived from existing measurement data. By advancing the window frame to incorporate new measurement data (e.g., one day data), short-term PMF analysis can provide source contributions for the most recent observations. This approach significantly reduces the time lag associated with receptor modelling source apportionment techniques. Such advancements hold important policy implications, as they enable prompt response during pollution episodes, eliminating the need to wait for the accumulation of sufficient data for conducting PMF analysis.

*Data availability.* Bihourly organic markers and other PM chemical speciation data presented in this study can be requested by contacting the corresponding author (jian.yu@ust.hk).

*Author contribution.* QW and JZY formulated the overall design of the study. QW, SZ, SW, CH, and YD carried out the measurement of PM chemical speciation data and data validation. QW did the overall data analysis with contributions from JZY. QW and JZY wrote the paper with contributions from all co-authors.

*Competing interests.* The authors declare that they have no conflict of interest.

*Acknowledgements.* We thank funding support from the Hong Kong Research Grants Council (R6011-18 and 16305418), the Hong Kong University of Science & Technology (VPRDO19IP01), the "CUG" Scientific Research Funds (No. 2022115) at China University of Geosciences (Wuhan), and the Fundamental Research Funds (No. G1323523063) for the Central Universities, China University of Geosciences (Wuhan).

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
