# Peer review of "Short-term Source Apportionment of Fine Particulate Matter with Time-dependent Profiles Using SoFi: Exploring the Reliability of Rolling Positive Matrix Factorization (PMF) Applied to Bihourly Molecular and Elemental Tracer Data"

_EGUsphere, 2023_

## Author Comment (AC1)

*Response to Review Comments by Anonymous Referee #1 on "Short-term Source Apportionment of Fine Particulate Matter with Time-dependent Profiles Using SoFi: Exploring the Reliability of Rolling Positive Matrix Factorization (PMF) Applied to Bihourly Molecular and Elemental Tracer Data" by Q. Wang et al.*

**General Comments by Anonymous Referee #1:**

I thoroughly reviewed this manuscript. The manuscript titled "Short-term Source Apportionment of Fine Particulate Matter with Time-dependent Profiles Using SoFi: Exploring the Reliability of Rolling Positive Matrix Factorization (PMF) Applied to Bihourly Molecular and Elemental Tracer Data" suggests that short-term PMF analysis with the *a*-value constraints in SoFi can be utilized to apportion primary sources accurately. This may have implications for short-term pollution episodes management. The topic is interesting and meaningful. However, there are some problems in the article. I would recommend authors concerns on the following comments. Minor revision is recommended.

**Response to General Comments:** We thank the reviewer for the comments and acknowledgment of the importance of our work. Below is our point-by-point response to each comment, marked in blue. The related text in the manuscript is copied here for reference, with newly added/revised text underlined.

Changes made to the main text will also be marked in blue in the revised manuscript file.

Line 35, "The implementation of stringent control measures since 2013 has led to declining concentrations of PM5 in many megacities in China", it is recommended to provide specific data on the decrease in PM2.5.

**Response:** Suggestion taken; the following statement will be added:

"The implementation of stringent control measures since 2013 has led to declining concentrations of $PM_{2.5}$ in many megacities in China, with annual-average decreased from 72.3 μg m$^{-3}$ in 2013 to 47.4 μg m$^{-3}$ in 2017, as calculated from measurements in 74 cities across China (Chow et al., 2022a; Wang et al., 2020)."

Line 55, "This limitation explains the common observation that PMF with robust mode tends to underestimate the high concentration data while overestimating the low concentration data.", it is recommended to modify this sentence to make it coherent with the preceding and following sentences.

**Response:** The reviewer may have misunderstood the following sentences. The sentence "Sporadic sources, such as firework emission…" mainly describe the pollution characteristics of the sporadic sources such as firework contributions, which can contribute significantly to PM pollution and offset the pollution controls of other anthropogenic sources. While the sentence "This limitation explains …"

describe the model performance of the PMF on the high conc. data, i.e., underestimation of the data, thus leading to the lower estimation of the corresponding sources. To improve the clarity, we will rephrase the sentences in the revised manuscript as following:

"Source profile changes over extended observation period are often expected for certain sources. For instance, biomass burning exhibits variations in dominant biomass materials across different seasons; the implementation of catalytic converter replacement program alters the source profiles of vehicular emissions (Lee et al., 2017). Sporadic sources, such as firework emission during holidays or wildfires during dry seasons, can significantly contribute to PM pollution episodes that persist for hours to days, often overshadowing the effects of reductions in anthropogenic emissions (Song et al., 2021; Kong et al., 2015). The PMF analysis using the long-term data sets could not properly reflect source profile changes experienced during the longtime span, which is commonly expected. In other words, long-term PMF is inherently unsuitable for apportioning sporadic sources or ephemeral pollution events. This limitation explains the common observation that PMF with robust mode tends to underestimate the high concentration data while overestimating the low concentration data (Henry and Christensen, 2010). Contribution estimates of these sources would be biased when apportioned alongside other regular sources using long-term observational data."

Line 99, please define SOA when it first appears in the text.

**Response:** We will revise it as suggested.

"…and α-pinene secondary organic aerosol (SOA) tracers …"

Line 121, it is recommended to explain what the meaning of k and j in Eq 4.

**Response:** We will revise it as suggested.

"…Here, the index $j$, which varies between 0 and the number of species-$m$, represents the species of the $k^{th}$ factor. The index $i$, which varies between 0 and the number of samples-$n$, is the sample of the $k^{th}$ factor. $f_{kj}$ and $g_{ik}$ are the anchoring profiles and anchoring contributions…"

Line 170, Do you think these "18 days" must be continuous? Does discontinuity have an impact on the results?

**Response:** No, the continuity is not a prerequisite. Hypothetically, there could be data gaps, as the individual sets of hourly observations are used as inputs and there is not a requirement that the individual observations should be continuous. Previous studies with higher time resolution measurement suggested a window length of 14 d, we did a preliminary test with our bihourly data sets and found that a longer time period would provide a more robust solution with cleaner source profiles.

We will revise the wording to improve the clarity in the revised manuscript:

"The short-term source apportionment analysis was conducted using data from the first sampling period, spanning 18 days from 29 Dec. 2019 to 15 Jan. 2020. The selection of the window length may vary depending on the specific data sets under study. The determination of the window length for our observational data set is shown in Text S2, where 4 d, 7 d, 10 d, 14 d and 18 d were initially evaluated."

Line 121, please define Q/Qexp.

**Response:** Suggestion taken; the following statement will be added:

"The ratio $Q/Q_{exp}$, where $Q_{exp} \approx n \times m$- $p \times (n + m)$, indicates the overall fitting of all input species and is reciprocally associated with the fitting (Norris et al., 2014)."

**Reference:**

Norris, G. A., Duvall, R., Brown, S. G., and Bai, S.: EPA Positive Matrix Factorization (PMF) 5.0 Fundamentals and User Guide, Environ. Prot. Agency Off. Researc Dev. Publushing House Whashington, DC 20460, 136, 2014.

Line 221, please explain what Nfirework_data

**Response:** Suggestion taken; the following statement will be added:

"…was excluded due to the limited availability of data points influenced by firework emissions ($N_{firework\_data}$=2, representing the number of data points under the influence of firework emissions)."

Line 240, please define

**Response:** Suggestion taken; the following statement will be added:

"Among the four secondary sources, secondary nitrate showed good agreement with the reference result (slope of 1.0 and Pearson correlation coefficient-$R_p$ of 1.0)."

Line 273, "Secondary nitrate, SOA_I and SOA_II showed large variations, with an average relative difference of 173%, 162%, and 75%, respectively.", it is recommended to provide a more detailed explanation of the reasons for the significant differences.

**Response:** We offer the following potential explanations for the significant differences. However, accurately attributing the contribution of each individual secondary source is challenging and requires additional measurement datasets to test and tune the methods.

The underlined text is newly added.

"Secondary nitrate, SOA_I and SOA_II showed large variations, with an average relative difference of 173%, 162%, and 75%, respectively. In the case of secondary nitrate factor, although the apportioned $PM_{2.5}$ contributions from individual source factors were comparable to the reference results, the resolved source profiles exhibited high time-dependent variabilities. We postulate this may be attributed to the sensitivity of nitrate formation to the reduction of $NO_x$ and VOC precursors during the lockdown restriction (Yang et al., 2022). Previous laboratory studies indicated that reducing anthropogenic pollutants such as $SO_2$ and $NO_x$ can also reduce the biogenic SOA formation via the anthropogenic–biogenic interaction (Zhang et al., 2019; Xu et al., 2015). This, to some extent, explains the high variabilities in source profiles of the two SOA factors. Additionally, the high variabilities may also arise from the uncertainties in the PMF analysis due to the limited data points available from the short-term

time span (Wang et al., 2018). Therefore, in future studies, alternative approaches are needed to independently assess the contribution of secondary sources. Also, we recommend deploying higher time resolution measurement of the organic tracers. This will help ensure accurate source apportionment results for individual secondary sources, especially within the confines of a short-term time span."

**References:**

Wang, Q., Qiao, L., Zhou, M., Zhu, S., Griffith, S., Li, L., and Yu, J. Z.: Source Apportionment of PM2.5 Using Hourly Measurements of Elemental Tracers and Major Constituents in an Urban Environment: Investigation of Time-Resolution Influence, J. Geophys. Res. Atmos., 123, 5284–5300, https://doi.org/10.1029/2017JD027877, 2018.

Xu, L., Guo, H., Boyd, C. M., Klein, M., Bougiatioti, A., Cerully, K. M., Hite, J. R., Isaacman-VanWertz, G., Kreisberg, N. M., Knote, C., Olson, K., Koss, A., Goldstein, A. H., Hering, S. V., De Gouw, J., Baumann, K., Lee, S. H., Nenes, A., Weber, R. J., and Ng, N. L.: Effects of anthropogenic emissions on aerosol formation from isoprene and monoterpenes in the southeastern United States, Proc. Natl. Acad. Sci. U. S. A., 112, 37–42, https://doi.org/10.1073/pnas.1417609112, 2015.

Zhang, Y. Q., Chen, D. H., Ding, X., Li, J., Zhang, T., Wang, J. Q., Cheng, Q., Jiang, H., Song, W., Ou, Y. B., Ye, P. L., Zhang, G., and Wang, X. M.: Impact of anthropogenic emissions on biogenic secondary organic aerosol: Observation in the Pearl River Delta, southern China, Atmos. Chem. Phys., 19, 14403–14415, https://doi.org/10.5194/acp-19-14403-2019, 2019.

Abstract and conclusion of the work can be improved as per the methods applied.

**Response:** We will revise the abstract and the conclusion to improve their readability and clarity in conveying the main findings in the revised manuscript.

---

## Author Response (AR1)

***Response to Review Comments by Anonymous Referee #1 on "Short-term Source Apportionment of Fine Particulate Matter with Time-dependent Profiles Using SoFi: Exploring the Reliability of Rolling Positive Matrix Factorization (PMF) Applied to Bihourly Molecular and Elemental Tracer Data" by Q. Wang et al.***

**General Comments by Anonymous Referee #1:**

I thoroughly reviewed this manuscript. The manuscript titled "Short-term Source Apportionment of Fine Particulate Matter with Time-dependent Profiles Using SoFi: Exploring the Reliability of Rolling Positive Matrix Factorization (PMF) Applied to Bihourly Molecular and Elemental Tracer Data" suggests that short-term PMF analysis with the *a*-value constraints in SoFi can be utilized to apportion primary sources accurately. This may have implications for short-term pollution episodes management. The topic is interesting and meaningful. However, there are some problems in the article. I would recommend authors concerns on the following comments. Minor revision is recommended.

**Response to General Comments:** We thank the reviewer for the comments and acknowledgment of the importance of our work. Below is our point-by-point response to each comment, marked in blue. The related text in the manuscript is copied here for reference, with newly added/revised text underlined. Changes made to the main text are marked in blue in the revised manuscript file.

Line 35, "The implementation of stringent control measures since 2013 has led to declining concentrations of PM5 in many megacities in China", it is recommended to provide specific data on the decrease in PM2.5.
**Response:** Suggestion taken; the following statement has been added:
**Lines 38-40:** "The implementation of stringent control measures since 2013 has led to declining concentrations of $PM_{2.5}$ in many megacities in China, with annual-average decreased from 72.3 μg m$^{-3}$ in 2013 to 47.4 μg m$^{-3}$ in 2017, as calculated from monitoring data in 74 cities across China (; Wang et al., 2020; Chow et al., 2022a)."

Line 55, "This limitation explains the common observation that PMF with robust mode tends to underestimate the high concentration data while overestimating the low concentration data.", it is recommended to modify this sentence to make it coherent with the preceding and following sentences.
**Response:** The reviewer may have misunderstood the following sentences. The sentence "Sporadic sources, such as firework emission…" mainly describes the pollution characteristics of the sporadic sources such as firework contributions, which can contribute significantly to PM pollution and offset the pollution controls of other anthropogenic sources. While the sentence "This limitation explains …" describes the model performance of the PMF on the high conc. data, i.e., underestimation of the data, thus leading to the lower estimation of the corresponding sources. To improve the clarity, we have rephrased the sentences in the revised manuscript as following:
**Lines 57-67:** "Source profile changes are often expected over an extended observation period for certain sources. For instance, biomass burning exhibits variations in dominant biomass materials during different seasons; the implementation of catalytic converter replacement program alters the source profiles of vehicular emissions (Lee et al., 2017). Sporadic sources, such as firework emission during holidays or wildfires during dry seasons, can significantly contribute to PM pollution episodes that persist for hours to days, often overshadowing the effects of reductions in anthropogenic emissions (Song et al., 2021; Kong et al., 2015). The PMF analysis using long-term data sets could not properly

reflect source profile changes experienced during the longtime span. In other words, long-term PMF is inherently unsuitable for apportioning sporadic sources or ephemeral pollution events. This limitation explains the common observation that PMF with robust mode tends to underestimate the high concentration data while overestimating the low concentration data (Henry and Christensen, 2010). Consequently, contribution estimates of these sources would be biased when apportioned alongside other regular sources using long-term observational data."

Line 99, please define SOA when it first appears in the text.
**Response:** We have revised it as suggested.
**Line 107:** "…and α-pinene secondary organic aerosol (SOA) tracers …"

Line 121, it is recommended to explain what the meaning of k and j in Eq 4.
**Response:** We have revised it as suggested.
**Lines 130-132:** "…Here, the index $j$, which varies between 0 and the number of species-$m$, represents the species of the $k^{th}$ factor. The index $i$, which varies between 0 and the number of samples-$n$, is the sample of the $k^{th}$ factor. $f_{kj}$ and $g_{ik}$ are the anchoring profiles and anchoring contributions…"

Line 170, Do you think these "18 days" must be continuous? Does discontinuity have an impact on the results?
**Response:** No, the continuity is not a prerequisite. Hypothetically, there could be data gaps, as the individual sets of hourly observations are used as inputs and there is not a requirement that the individual observations should be continuous. Previous studies with higher time resolution measurement suggested a window length of 14 d, we did a preliminary test with our bihourly data sets and found that a longer time period would provide a more robust solution with cleaner source profiles.

We have revised the wording to improve the clarity in the revised manuscript:

**Lines 195-198:** "The short-term source apportionment analysis was conducted using data from the first sampling period, spanning 18 days from 29 Dec. 2019 to 15 Jan. 2020. The selection of the window length may vary depending on the specific data sets under study. The determination of the window length for our observational data set is shown in Text S2, where 4 d, 7 d, 10 d, 14 d and 18 d were initially evaluated."

Line 121, please define Q/Qexp.
**Response:** Suggestion taken; the following statement has been added:

**Lines 209-210:** "The ratio $Q/Q_{exp}$, where $Q_{exp} \approx n \times m - p \times (n + m)$, indicates the overall fitting of all input species and is reciprocally associated with the fitting (Norris et al., 2014)."
**Reference:**
Norris, G. A., Duvall, R., Brown, S. G., and Bai, S.: EPA Positive Matrix Factorization (PMF) 5.0 Fundamentals and User Guide, Environ. Prot. Agency Off. Researc Dev. Publushing House Whashington, DC 20460, 136, 2014.

Line 221, please explain what Nfirework_data
**Response:** Suggestion taken; the following statement has been added:

**Lines 252-254:** "…was excluded due to the limited availability of data points influenced by firework emissions ($N_{\text{firework\_data}}=2$, representing the number of data points under the influence of firework

emissions).”

Line 240, please define

**Response:** Suggestion taken; the following statement has been added:

**Lines 274-275: “**Among the four secondary sources, secondary nitrate showed good agreement with the reference result (slope of 1.0 and Pearson correlation coefficient-$R_p$ of 1.0).”

Line 273, “Secondary nitrate, SOA_I and SOA_II showed large variations, with an average relative difference of 173%, 162%, and 75%, respectively.”, it is recommended to provide a more detailed explanation of the reasons for the significant differences.

**Response:** We have offered the following potential explanations for the significant differences in the revised manuscript. However, accurately attributing the contribution of each individual secondary source is challenging and requires additional measurement datasets to test and tune the methods.

The underlined text is newly added.

**Lines 308-319: “**Secondary nitrate, SOA_I and SOA_II showed large variations, with an average relative difference of 173%, 162%, and 75%, respectively. In the case of secondary nitrate factor, although the apportioned $PM_{2.5}$ contributions from individual source factors were comparable to the reference results, the resolved source profiles exhibited high time-dependent variabilities. We postulate this may be attributed to the sensitivity of nitrate formation to the reduction of $NO_x$ and VOC precursors during the lockdown restriction (Yang et al., 2022). Previous laboratory studies indicated that reducing anthropogenic pollutants such as $SO_2$ and $NO_x$ can also reduce the biogenic SOA formation via anthropogenic–biogenic interactions (Zhang et al., 2019; Xu et al., 2015). This, to some extent, explains the high variabilities in source profiles of the two SOA factors. Additionally, the high variabilities may also arise from the uncertainties in the PMF analysis due to the limited data points available from the short-term time span (Wang et al., 2018). Therefore, in future studies, alternative approaches are needed to independently assess the contribution of secondary sources. Also, we recommend deploying higher time resolution measurement of the organic tracers. This will help ensure accurate source apportionment results for individual secondary sources, especially within the confines of a short-term time span.”

**References:**

Wang, Q., Qiao, L., Zhou, M., Zhu, S., Griffith, S., Li, L., and Yu, J. Z.: Source Apportionment of PM2.5 Using Hourly Measurements of Elemental Tracers and Major Constituents in an Urban Environment: Investigation of Time-Resolution Influence, J. Geophys. Res. Atmos., 123, 5284–5300, https://doi.org/10.1029/2017JD027877, 2018.

Xu, L., Guo, H., Boyd, C. M., Klein, M., Bougiatioti, A., Cerully, K. M., Hite, J. R., Isaacman-VanWertz, G., Kreisberg, N. M., Knote, C., Olson, K., Koss, A., Goldstein, A. H., Hering, S. V., De Gouw, J., Baumann, K., Lee, S. H., Nenes, A., Weber, R. J., and Ng, N. L.: Effects of anthropogenic emissions on aerosol formation from isoprene and monoterpenes in the southeastern United States, Proc. Natl. Acad. Sci. U. S. A., 112, 37–42, https://doi.org/10.1073/pnas.1417609112, 2015.

Zhang, Y. Q., Chen, D. H., Ding, X., Li, J., Zhang, T., Wang, J. Q., Cheng, Q., Jiang, H., Song, W., Ou, Y. B., Ye, P. L., Zhang, G., and Wang, X. M.: Impact of anthropogenic emissions on biogenic secondary organic aerosol: Observation in the Pearl River Delta, southern China, Atmos. Chem. Phys., 19, 14403–14415, https://doi.org/10.5194/acp-19-14403-2019, 2019.

Abstract and conclusion of the work can be improved as per the methods applied.

**Response:** We have revised the abstract and the conclusion to improve their readability and clarity in conveying the main findings and their significance.

**Lines 30-33: "**…, Secondary while future efforts are needed to improve the prediction of individual secondary sources. Additionally, future rapid source apportionment analysis can benefit from utilizing a library of source profiles derived from existing measurement data, thereby significantly reducing the time lag associated with receptor modelling source apportionment techniques."

**Lines 328-349: "**In this study, we presented a short-term PMF strategy utilizing bihourly PM chemical speciation data including the molecular and elemental tracers. Initially, the $PMF_{ref}$ using the campaign-wide measurement data were performed by EPA PMF software to get an overview of the emission sources and obtain the reference profiles of the primary sources. Then, the short-term PMF analysis was performed. … Nevertheless, the overall contribution of the total secondary sources showed good agreement. Future endeavors should target to improve the modelling of individual secondary factors by either using alternative approaches or deploying higher time resolution measurement of organic tracers.

The findings of this study highlight the applicability of the short-term PMF analysis with source profile constraints for source apportionment of $PM_{2.5}$. This suggests the potential for future work to achieve rapid source apportionment by utilizing a library of source profiles derived from existing measurement data. By advancing the window frame to incorporate new measurement data (e.g., one day data), short-term PMF analysis can provide source contributions for the most recent observations. This approach significantly reduces the time lag associated with receptor modelling source apportionment techniques. Such advancements hold important policy implications, as they enable prompt response during pollution episodes, eliminating the need to wait for the accumulation of sufficient data for conducting PMF analysis."

***Response to Review Comments by Anonymous Referee #2 on "Short-term Source Apportionment of Fine Particulate Matter with Time-dependent Profiles Using SoFi: Exploring the Reliability of Rolling Positive Matrix Factorization (PMF) Applied to Bihourly Molecular and Elemental Tracer Data" by Q. Wang et al.***

**General Comments by Anonymous Referee #2:**

The manuscript titled "Short-term Source Apportionment of Fine Particulate Matter with Time-dependent Profiles Using SoFi: Exploring the Reliability of Rolling Positive Matrix Factorization (PMF) Applied to Bihourly Molecular and Elemental Tracer Data" presents extensive datasets of real-time chemical characterization to employ both traditional and rolling Positive Matrix Factorization (PMF) techniques utilizing the SoFi software. The comparative analysis between traditional and rolling PMF methodologies is particularly intriguing, primarily in the context of effectively modeling primary sources, which exhibit relatively minor variabilities. Conversely, the study reveals substantial variations in the case of secondary factors. This result deserves more explanation and details regarding the relative

differences. Furthermore, from what has been presented in the paper, the rolling PMF should always be based on source profiles from traditional PMF (Conclusion lines 300-302). This prompts the question of how to address scenarios where no source profiles are available, such as for newly emerging sources or in regions lacking local source profiles.

I recommend major revisions before considering the manuscript for acceptance:

**Response to General Comments:** We thank the reviewer for the comments. Below is our point-by-point response to each comment, marked in blue. The related text in the manuscript is copied here for reference, with newly added/revised text underlined. Changes made to the main text will also be marked in blue in the revised manuscript file.

(1) Regarding the substantial variations in secondary factors, please see our response to a similar comment by Reviewer #1. The response text is copied below for easy reference. The underlined text is newly added.

**Lines 308-319:** "Secondary nitrate, SOA_I and SOA_II showed large variations, with an average relative difference of 173%, 162%, and 75%, respectively. In the case of secondary nitrate factor, although the apportioned PM$_{2.5}$ contributions from individual source factors were comparable to the reference result, the resolved source profiles exhibited high time-dependent variabilities. We postulate this may be attributed to the sensitivity of nitrate formation to the reduction of NO$_x$ and VOC precursors during the lockdown restriction (Yang et al., 2022). Previous laboratory studies indicated that reducing anthropogenic pollutants such as SO$_2$ and NO$_x$ can also reduce the biogenic SOA formation via anthropogenic–biogenic interactions (Zhang et al., 2019; Xu et al., 2015). This, to some extent, explains the high variabilities in source profiles of the two SOA factors. Additionally, the high variabilities may also arise from the uncertainties in the PMF analysis due to the limited data points available from the short-term time span (Wang et al., 2018). Therefore, in future studies, alternative approaches are needed to independently assess the contribution of secondary sources. Also, we recommend deploying higher time resolution measurement of the organic tracers. This will help ensure accurate source apportionment results for individual secondary sources, especially within the confines of a short-term time span."

**References:**

Wang, Q., Qiao, L., Zhou, M., Zhu, S., Griffith, S., Li, L., and Yu, J. Z.: Source Apportionment of PM2.5 Using Hourly Measurements of Elemental Tracers and Major Constituents in an Urban Environment: Investigation of Time-Resolution Influence, J. Geophys. Res. Atmos., 123, 5284–5300, https://doi.org/10.1029/2017JD027877, 2018.

Xu, L., Guo, H., Boyd, C. M., Klein, M., Bougiatioti, A., Cerully, K. M., Hite, J. R., Isaacman-VanWertz, G., Kreisberg, N. M., Knote, C., Olson, K., Koss, A., Goldstein, A. H., Hering, S. V., De Gouw, J., Baumann, K., Lee, S. H., Nenes, A., Weber, R. J., and Ng, N. L.: Effects of anthropogenic emissions on aerosol formation from isoprene and monoterpenes in the southeastern United States, Proc. Natl. Acad. Sci. U. S. A., 112, 37–42, https://doi.org/10.1073/pnas.1417609112, 2015.

Zhang, Y. Q., Chen, D. H., Ding, X., Li, J., Zhang, T., Wang, J. Q., Cheng, Q., Jiang, H., Song, W., Ou, Y. B., Ye, P. L., Zhang, G., and Wang, X. M.: Impact of anthropogenic emissions on biogenic secondary organic aerosol: Observation in the Pearl River Delta, southern China, Atmos. Chem. Phys., 19, 14403–14415, https://doi.org/10.5194/acp-19-14403-2019, 2019.

(2) Regarding how to address scenarios where no source profiles are available, we'd like to note that the methodology presented here requires pre-existing monitoring compositional data at a locality that allows initial PMF analysis, which would generate a set of source profiles. For newly emerging sources, as long as we captured the event using the high time resolution measurement data, we can still build its local source profiles based on the measurement data. Then the source profiles can be documented and used later in the short-term source apportionment analysis for this location. We would like to emphasize that the purpose of such short-term source apportionment is to achieve the rapid source apportionment analysis to better serve policymaking. This point is elaborated in the conclusion section and copied below:

**Lines 344-349:** "This suggests the potential for future work to achieve rapid source apportionment by utilizing a library of source profiles derived from existing measurement data. By advancing the window frame to incorporate new measurement data (e.g., one day data), short-term PMF analysis can provide source contributions for the most recent observations. This approach significantly reduces the time lag associated with receptor modelling source apportionment techniques. Such advancements hold important policy implications, as they enable prompt response during pollution episodes, eliminating the need to wait for the accumulation of sufficient data for conducting PMF analysis."

Introduction: It is imperative to clarify that the traditional PMF was conducted using the US EPA PMF, as indicated in your prior publication (Wang et al., 2022b). This should be explicitly mentioned unless this is not the case.
**Response:** The following underlined text are newly added and have been included in the revised manuscript:

**Lines 51-53:** "Due to the limited time resolution from offline filter-based sampling schedule, e.g., sampling duration of 24 h and sampling frequency of once every three or six days, PMF is often conducted using the Environmental Protection Agency-EPA PMF software (Norris et al., 2014) with data spanning one or multi years to meet the sample size requirement (e.g., Chow et al., 2022b; Scotto et al., 2021)."

**Lines 92-95:** "A thorough traditional source apportionment analysis conducted using the EPA PMF software is documented in our previous study (Wang et al., 2022b). In this study, we specifically investigated the applicability of a short-term source apportionment strategy using the bihourly $PM_{2.5}$ chemical speciation data with the SoFi software and compared with those obtained through the traditional PMF".

Section 2.2: This section is lacking essential details regarding the specific species incorporated into the model, the total number of data utilized for traditional and rolling PMF, and the temporal resolution applied (e.g., 2-hour or 1-hour intervals). These specifics are fundamental for a comprehensive understanding of the methodology.
**Response:** The relevant information is provided below and have been added in this section in the revised manuscript:

**Lines 136-145:** "Figure 1 illustrates the flowchart outlining the source apportionment methodology employed in this study. Initially, a PMF run was conducted by EPA PMF software using campaign-wide bihourly data as input (referred to as $PMF_{ref}$) to derive the reference profiles for primary sources.…The results of the rolling PMF analysis were discussed and compared with the results obtained from $PMF_{ref}$. The 22 input species for both the $PMF_{ref}$ and the short-term rolling PMF runs

include sulfate, nitrate, ammonium, OC, EC, K, Ca, Mn, Fe, Cu, Zn, As, Ba, Pb, and 8 organic species (hopanes, steranes, levoglucosan, mannosan, phthalic acid, α-pinT, β-caryT, and DHOPA). The specific input data utilized in individual PMF runs are shown in Table S1."

**Table S1. Summary of the input data utilized in the reference run (PMF$_{ref}$), short-term PMF testing run and rolling PMF runs (PMF$_{roll}$) using the remaining data performed in this study.**

|  | Run No. | Starting Time | Ending Time | Input sample size | Factor numbers |
|---|---|---|---|---|---|
| PMF$_{ref}$ run |  | 0:00 29 Dec. 2019 | 22:00 9 Feb. 2020 | 416 | 10 |
| Short-term PMF run |  | 0:00 29 Dec. 2019 | 16:00 15 Jan. 2020 | 190 | 9 |
| PMF$_{roll}$ with remaining dataset | 1 | 0:00 30 Dec. 2019 | 22:00 22 Jan. 2020 | 195 | 9 |
|  | 2 | 0:00 31 Dec. 2019 | 22:00 23 Jan. 2020 | 193 |  |
|  | 3* | 0:00 1 Jan. 2020 | 22:00 24 Jan. 2020 | 193 |  |
|  | 4 | 0:00 2 Jan. 2020 | 20:00 25 Jan. 2020 | 196 |  |
|  | 5 | 0:00 3 Jan. 2020 | 22:00 26 Jan. 2020 | 196 |  |
|  | 6 | 0:00 4 Jan. 2020 | 20:00 27 Jan. 2020 | 196 |  |
|  | 7 | 0:00 5 Jan. 2020 | 22:00 28 Jan. 2020 | 196 |  |
|  | 8 | 0:00 6 Jan. 2020 | 22:00 29 Jan. 2020 | 197 |  |
|  | 9 | 16:00 7 Jan. 2020 | 20:00 30 Jan. 2020 | 196 |  |
|  | 10 | 0:00 8 Jan. 2020 | 22:00 31 Jan. 2020 | 204 |  |
|  | 11 | 0:00 9 Jan. 2020 | 20:00 1 Feb. 2020 | 206 | 10 |
|  | 12 | 0:00 10 Jan. 2020 | 22:00 2 Feb. 2020 | 209 |  |
|  | 13 | 0:00 11 Jan. 2020 | 20:00 3 Feb. 2020 | 208 |  |
|  | 14 | 0:00 12 Jan. 2020 | 22:00 4 Feb. 2020 | 209 |  |
|  | 15 | 0:00 13 Jan. 2020 | 22:00 5 Feb. 2020 | 209 |  |
|  | 16 | 0:00 14 Jan. 2020 | 20:00 6 Feb. 2020 | 208 |  |
|  | 17 | 0:00 15 Jan. 2020 | 22:00 7 Feb. 2020 | 208 |  |
|  | 18 | 0:00 22 Jan. 2020 | 22:00 8 Feb. 2020 | 208 |  |
|  | 19 | 0:00 23 Jan. 2020 | 22:00 9 Feb. 2020 | 208 |  |

*Run No. 3 was excluded due to very limited firework-influence data point in the input samples, leading to outlier results compared with other rolling PMF runs.

Line 128: The choice of an 18-day duration for data inclusion needs further justification. Is this duration adequate for achieving robust and optimal *a*-values?

**Response:** The justification of the selection of the duration of the data are provided in Text S2, where bootstrap error estimation and factor profile mixing were examined. Compared with the shorter time window, the 18 d results showed less factor mixing. The results also passed the bootstrap error estimation. We have refined the text in the revised manuscript to improve the clarity:

**Lines 195-198: "**The short-term source apportionment analysis was conducted using data from the first sampling period, spanning 18 days from 29 Dec. 2019 to 15 Jan. 2020. The selection of the window length may vary depending on the specific data sets under study. The determination of the window length for our observational data set is shown in Text S2, where 4 d, 7 d, 10 d, 14 d and 18 d were initially evaluated."

Line 147: The difference in the correlation is clear for K and levoglucosan but what about K and Pb? They're still well correlated even in the CNY period?

**Response:** The good correlation between K and Pb may indicate the common regional origins of the two sources (i.e., biomass burning and coal combustion) at our observational site. Firework emissions emit extra amount of K compared with coal combustion, which leads to the elevated slope of K vs. Pb during the CNY period in Figure 2b.

Line 150: A discrepancy is noted between the number of factors reported in the paper by Wang et al., 2022b (14 factors) and the present manuscript (10 factors). The differences in PMF methodologies should be thoroughly explained and supported with additional details, either integrated into the main text or provided as supplementary information. Additionally, the manuscript should incorporate validation results for traditional PMF, including Bootstrap analysis, DISP analysis, reconstruction of species, Q/Qexp...

**Response:** We have provided following additional details about the PMF$_{ref}$ runs in the revised manuscript:

**Lines 161-173:** "Source apportionment results over entire sampling period (i.e., PMF$_{ref}$) supplies an overview about the emission sources at this site. A thorough source apportionment result for this site can be found in our previous paper (Wang et al., 2022b), where 14 factors were resolved using a list of more comprehensive input species over the entire sampling period. Among these factors, the PAH-rich factor, cooking emission, and one SOA factor are negligible PM$_{2.5}$ contributors (<1%). The contribution of the residual oil combustion factor to PM$_{2.5}$ is also minor (<3%). Additionally, the detection frequency of V, a tracer for the residual oil combustion factor, was lower than 50% for the short-term input time window. Thus, these four factors were not incorporated in this study, and we focus on the 10 major factors resolved in our PMF$_{ref}$ run. Using the data points falling in the short-term PMF runs, the 10-factor solution is more robust. The robustness of the PMF$_{ref}$ result was tested by the bootstrap and displacement error estimation method embedded in EPA PMF 5.0 software (Norris et al., 2014). All bootstrap factors mapped to the base factors in >95% of the runs. No factor swaps and no decrease of Q were observed in the displacement result. The PMF-modelled reconstructed PM$_{2.5}$ mass is close to the measured one, with slope of 1.01 and R$_p$ of 0.99. The model performance for individual species was also good, with slopes ranged from 0.59 to 1.08 and R$_p$ in the range of 0.82-1.00."

Section 3.2: What is the reason behind not adding an *a*-value to the secondary sources?

**Response:** The reasons are multi-fold. First, compared with the primary sources, the secondary sources often do not represent specific emissions. Instead, they typically result from a complex interplay of multiple aging processes that occur over the measurement period. Second, the secondary processes are often heavily influenced by environmental conditions at a given locality, including factors such as relative humidity and photoactivity. Thus, the potential variabilities of secondary sources would be larger, resulting in higher uncertainties for estimating secondary sources. The secondary sources are left unconstrained as well in previous studies (Chen et al., 2022; Canonaco et al., 2013).

The revised text is copied here for reference, and have been incorporated in the revised manuscript:

**Lines 204-208:** "Compared with primary sources, the secondary sources often do not represent specific emissions. Instead, they typically result from a complex interplay of multiple aging processes that occur over the observational period and are susceptible to environmental conditions such as relative humidity and photoactivity, etc. Thus, the four secondary factors were not constrained in the short-term PMF run using the *a*-value approach, consistent with the common strategy in previous studies (Chen et al., 2022; Canonaco et al., 2021)."

Section 3.4: The variabilities observed in secondary profiles demand a more in-depth exploration. Possible explanations, including the potential impact of poorly resolved profiles in traditional PMF, should be examined and discussed in greater detail.
**Response:** We agree that the high variabilities of the secondary sources need more in-depth exploration. Reviewer #1 raises the same comment. Please see the detailed response text at the beginning of this document where we address the general comments by the reviewer.

In the SoFi software, what are the validation methods in order to ensure that you have good results?
**Response:** Currently, the bootstrap resampling strategy is not available in the standard-unlimited version of SoFi. In order to get a robust solution, we increased the number of PMF calls to 100 to ensure the minimum $Q$ value obtained. Among the 100 runs, the obtained $Q$ values are quite stable.

The following text have been added in the revised manuscript:

**Lines 209-212:** "A total of 100 PMF calls were performed and the variability of the $Q/Q_{exp}$ was examined. The ratio $Q/Q_{exp}$, where $Q_{exp} \approx n \times m - p \times (n + m)$, indicates the overall fitting of all input species and is reciprocally associated with the fitting (Norris et al., 2014). Among the 100 runs, the variation of the $Q/Q_{exp}$ are consistently minimum, with a coefficient of variation of <1%. The one with the lowest $Q/Q_{exp}$ was chosen for further analysis."